# Functional Dynamics of Substrate Recognition in TEM Beta-Lactamase

**DOI:** 10.3390/e24050729

**Published:** 2022-05-20

**Authors:** Chris Avery, Lonnie Baker, Donald J. Jacobs

**Affiliations:** 1Department of Bioinformatics and Genomics, University of North Carolina at Charlotte, Charlotte, NC 28223, USA; cavery12@uncc.edu (C.A.); lbaker46@uncc.edu (L.B.); 2Department of Physics and Optical Science, University of North Carolina at Charlotte, Charlotte, NC 28223, USA

**Keywords:** beta-lactamase, beta-lactam antibiotics, molecular dynamics, dynamic allostery, functional dynamics, machine learning, SPLOC

## Abstract

The beta-lactamase enzyme provides effective resistance to beta-lactam antibiotics due to substrate recognition controlled by point mutations. Recently, extended-spectrum and inhibitor-resistant mutants have become a global health problem. Here, the functional dynamics that control substrate recognition in TEM beta-lactamase are investigated using all-atom molecular dynamics simulations. Comparisons are made between wild-type TEM-1 and TEM-2 and the extended-spectrum mutants TEM-10 and TEM-52, both in apo form and in complex with four different antibiotics (ampicillin, amoxicillin, cefotaxime and ceftazidime). Dynamic allostery is predicted based on a quasi-harmonic normal mode analysis using a perturbation scan. An allosteric mechanism known to inhibit enzymatic function in TEM beta-lactamase is identified, along with other allosteric binding targets. Mechanisms for substrate recognition are elucidated using multivariate comparative analysis of molecular dynamics trajectories to identify changes in dynamics resulting from point mutations and ligand binding, and the conserved dynamics, which are functionally important, are extracted as well. The results suggest that the H10-H11 loop (residues 214-221) is a secondary anchor for larger extended spectrum ligands, while the H9-H10 loop (residues 194-202) is distal from the active site and stabilizes the protein against structural changes. These secondary non-catalytically-active loops offer attractive targets for novel noncompetitive inhibitors of TEM beta-lactamase.

## 1. Introduction

Drug resistance in bacteria remains a major concern for public health [1,2]. Antibiotics have been the tool of choice in the fight against bacterial infections following their inception with penicillin. Due to widespread overuse and poor waste management, resistance mechanisms have spread throughout the global bacterial community, rendering many drugs clinically ineffective [3,4]. Despite increasing resistance, beta-lactams remain the most commonly prescribed class of antibiotics [5]. Beta-lactams are small molecules that contain the 4-membered beta-lactam ring motif, and include penicillin, cephalosporin, monobactam, and carbapenem-like molecules [6]. These drugs act by binding to transpeptidase (penicillin binding protein), preventing them from linking together the peptidoglycans which hold a bacteria’s cell wall together, ultimately leading to bacteria cell death [7].

One of the leading causes of resistance to beta-lactams in bacteria is the production of an enzyme called beta-lactamase [8]. Beta-lactamase proteins are broadly categorized into four major classes, namely, A, B, C, and D, based on structural similarity [9]. Classes A, C, and D share the same serine-mediated catalytic mechanism for inactivating beta-lactams, while class B is a family of metalloenzymes that use a unique zinc-mediated hydrolysis mechanism. The classes are further divided into families with a high sequence identity. One of the earliest class A beta-lactamase enzymes to be encountered was TEM-1, which is considered the wild-type variant [10]. In recent decades many new TEM beta-lactamases have been found worldwide, partly due to new antibiotics that induce adaptations in beta-lactamase enzymes to protect their parent bacteria.

Alternatively, beta-lactamases can be classified based on the class of drugs they provide resistance to [11]. Narrow Spectrum Beta-Lactamase (NSBL) primarily exhibits penicilinase and limited cephalosporinase activity, which is considered the wild-type substrate profile (Bush-Jacoby classification 2a, 2b). Additional classes of beta-lactamase include Extended Spectrum Beta-Lactamase (ESBL), which confers resistance against third-generation cephalosporins and beyond (2be, 2de, 2e, 2ber) [12]; inhibitor-resistant beta-lactamase (IRBL) [13], able to confer resistance in the presence of traditional beta-lactamase inhibitors (2br, 2ber); and more recently, carbapenemase beta-lactamase (CBL) [14], which today is the main source of concern regarding beta-lactamase-mediated antibiotic resistance (2f, 3a, 3b).

Outside of its clinical significance, beta-lactamase provides an interesting model for studying the role of dynamics on ligand binding. Despite hydrolysis of many substrates, the catalytic mechanism is relatively conserved between class A, C, and D enzymes. The structure of wild-type TEM-1 beta-lactamase is shown in Figure 1, with the active site highlighted in magenta. The core residues responsible for hydrolysis are Ser70 (the primary catalytic residue), Lys73, Ser130, Asn132, Glu166, Lys234, and Ala237 [15]. Ser70 is activated by Lys73 and covalently bonds to the beta-lactam ring, while Glu166 activates a catalytic water to deacylate this bond. The result is that the beta-lactam ring is broken, leaving the drug inactive. Other catalytic residues help transport protons for hydrolysis and maintain the structure of the catalytically active site [16]. As this mechanism is well conserved, mutations outside the catalytic pocket control substrate specificity.

Mutations in TEM beta-lactamase at residues such as Arg164 and Gly238, known as primary mutations, are often associated with novel resistance through the reduction of substrate specificity [15]. Many secondary mutations can fine-tune resistance spectra, and mutations far from the active site (such as residue Met182) support primary mutations by increasing the thermodynamic stability of the enzyme. Many mutations have no or unknown effects on the function of beta-lactamase [17], such as Gln39Lys, which distinguishes wild-type TEM-1 and TEM-2 [18,19].

In general, the effects of a mutation may induce changes in structure or motions far away from the vicinity of the substitution [20]. Mutations can impact binding efficacy by changing how different substrates are accommodated into the binding pocket by inhibiting or enhancing functionality related to changes in active site dynamics. Additionally, dynamic allostery can modulate binding affinities after the binding of a secondary ligand or cofactor at another binding site perturbs the dynamics in the active site [21].

In this work, the dynamics of four class A TEM beta-lactamases are investigated by Molecular Dynamics (MD) simulation. The dataset contains simulations of TEM-1 and TEM-2 (representing NSBL) and TEM-10 and TEM-52 (representing ESBL) both in apo form and bound to four antibiotic substrates: ampicillin, amoxicillin, cefotaxime, and ceftazidime. The functional dynamics critical for substrate recognition are investigated using a model of dynamic allostery developed via machine learning. Dynamic allostery is a form of functional dynamics that deals with how a secondary binding event that is distant from the catalytic pocket alters vibrations [22]. Mechanistic explanations involving correlated motions are identified through a multivariate comparative analysis of molecular dynamics trajectories using Supervised Projective Learning for Orthogonal Completeness (SPLOC).

In the rest of this paper, Section 2 describes the MD simulations, the model for predicting dynamic allostery, and how functional dynamics were extracted using the SPLOC machine learning method. Section 3 describes the results of the MD simulations by analyzing dynamic allostery and functional dynamics. Section 4 explains the significance of the results and how the results of this work are consistent with other previously reported results in the literature. Finally, Section 5 concludes by highlighting the successes of the methods used to gain insight into the dynamic mechanisms of substrate binding, and proposes a strategy for novel inhibitors that utilize long-range dynamic allostery.

## 2. Materials and Methods

### 2.1. System Preparation

Eight X-ray crystal structures (PDB codes: 1ERM, 1ERO, 1ERQ, 1HTZ, 1JWP, 1LHY, 1XPB, and 3JYI [23,24,25,26,27,28]) of TEM beta-lactamases were collected and downloaded from the Protein Data Bank [29]. The all-atom structure of each enzyme was extracted by removing any crystallization waters, co-factors, or ligands. In pymol [30], each crystal structure was mutated to represent TEM-1, and then further mutated so that each of the eight crystal structures had a structure representing each of the four TEM mutants: TEM-1 (wild type), TEM-2 (Gln39Lys), TEM-10 (Arg164Ser/Glu240Lys), and TEM-52 (Glu104Lys/Met182Thr/Gly238Ser). In total this resulted in 32 starting apo structures.

For the ligands, the PDB structures (PDB codes: 3KP3, 6I1E, 4PM5, and 5TWE [31,32,33,34]) provided an example of each of the ligands in complex with beta-lactamase or another enzyme. The coordinates of the all-atom models for each ligand were extracted and used as a base model. When necessary, bonds were reformed in Avogadro [35] to represent the intact ligand prior to enzyme hydrolysis. Special care ensured that each molecule was in its bioactive protonation state [36], as shown in Figure 2.

Four ligands were docked into the binding pocket of a beta-lactamase structure; ampicillin and amoxicillin were docked with an apo TEM-1 molecule (1XPB) and cefotaxime and ceftazidime were docked with an apo TEM-52 molecule (1HTZ), following the functional substrate profiles of the enzymes. Rigid receptor docking was performed using AutoDock Vina [37] and DockThor [38,39], and the resulting top-scoring poses were examined for physical relevance. In particular, the beta lactam ring was positioned in the active site such that interactions with Ser70 were favorable. In order to obtain the sixteen starting structures for MD simulation, the four previously-obtained docked structures were used as templates to align each of the four beta-lactamase mutants with the four ligand types by superposition; this methodology was chosen because all of the crystal structures had high structural similarity (RMSD < 0.35 Å). Each ligand was then observed in order to determine how it naturally reacted to the presence of the beta-lactamase active site and mutations.

### 2.2. Molecular Dynamics Simulation

Simulations were performed following the protocol described in prior work [40]. The amber99sb-ildn protein force field [41] was used to compute molecular interactions at the all-atom level. Each ligand was parameterized using Antechamber [42] and ACPYPE [43] was used to convert the ligand parameters to the GROMACS file format, which were then added to the itp and topology files. All simulations were performed using GROMACS [44]. For each system, the starting structure was placed in a cubic simulation box with 1 nm distance between the edge of the protein and the side of the box, then solvated in TIP3P water solvent. Sodium ions were added in order to neutralize the net electrostatic charge of the system. Energy minimization was applied until the net force on any atom did not exceed 1000 kJ/(mol nm). The system was then coupled to a Berendenson thermostat and equilibrated to 300 K for 1 ns, then to a Parrinello–Rahman barostat and equilibrated to 1 bar for another 1 ns, followed by a 500 ns production run. The first 100 ns in the production run for each mutant was used for further equilibration of the protein.

There are 32 apo systems, consisting of the TEM-1, TEM-2, TEM-10 and TEM-52 mutants, each starting with eight different crystal structures, yielding a combined total of 16 μs of dynamics. There are 16 holo systems where ampicillin, amoxicillin, cefotaxime, and ceftazidime are bound to mutants TEM-1, TEM-2, TEM-10, and TEM-52 using crystal structures 1XPB and 1HTZ for broad and extended spectrum antibiotics, respectively. For holo systems, a combined total of 8 μs of dynamics was simulated. In all cases, the trajectories were centered in the simulation box, and no nonphysical artifacts appeared when visualizing the motion.

Using Java Essential Dynamics Inspector (JEDi) [45], each trajectory was aligned to a common reference frame (the first frame of the 1ERM TEM-1 simulation) and the carbon alpha coordinates were extracted into a plain text file that could be read into other analysis programs. Further validation for convergence in the MD trajectories was performed by computing the RMSD and essential dynamics for the simulations. RMSD was calculated for each trajectory according to the equation
(1)$$RMSD\left(t\right)=\sqrt{<{\left(\vec{x_{i}}\left(t\right)-{\vec{x}}_{ref}\right)}^{2}{>}_{i}}$$
where ${\vec{x}}_{ref}$ represents the alpha carbon coordinates of the reference conformation. The RMSD as a function of time shows how the global motion of an enzyme fluctuates during a simulation. The essential dynamics of the apo and holo trajectories were found by constructing and diagonalizing the pooled covariance matrix of all MD simulations. The distributions of the first two principal components were then analyzed using a PDF estimator in MATLAB [46,47].

### 2.3. Dynamic Allostery Model

Dynamic allostery is quantified by observing changes in correlated motions due to coupled binding events. Normal modes with the lowest frequencies represent the largest-scale correlated motions of a protein, which contribute most to the transmission of dynamic allostery. Under the quasi-harmonic approximation, the unperturbed Hessian matrix $H_{o}$ is proportional to the inverse of the covariance matrix Σ [48]. From an MD trajectory of an apo mutant, the unperturbed Hessian matrix is provided by
(2)$$H_{o}=RT\;\Sigma^{-1}=RT\sum_{k=1}^{p}k\frac{1}{\lambda_{k}}k$$
where $\lambda_{k}$ is the eigenvalue for the *k*-th eigenvector, |k〉, of the covariance matrix, and *p* is the number of eigenvectors (modes) equal to the state space dimension.

The modes with the lowest eigenvalues of the covariance matrix dominate the sum in Equation (Equation 2), yet they characterize noise with a small amplitude motion. In previous work [49] a sound approach to regularizing the process of taking the inverse of the covariance matrix was provided. However, the method employed here removes the previously-introduced hyper-parameter and shifts the spectrum of the covariance matrix to a much lesser degree than described earlier [49]. In order to decorrelate noise in the covariance matrix, all eigenvalues that fall below a minimum value, $\lambda_{m}$, are replaced by this minimum value using the floor operation $\lambda_{k}\to \max\left(\lambda_{k},\lambda_{m}\right)$ prior to using Equation (Equation 2). This minimum value corresponds to the square of the estimated uncertainty in the atomic coordinates. Here, the minimum uncertainty in the atomic coordinates is estimated at 0.01 Å.

The effect of ligand binding is modeled by adding pairwise harmonic restraints (springs) in a local spherical region that encapsulates a binding site. Each spring with spring constant $k_{s}$ modifies the unperturbed Hessian matrix. The model used here deals only with carbon alpha atoms. The region is defined as connecting any carbon alpha atom within 10 Å from a tagged carbon alpha atom that is a member of the binding site. Under quasi-harmonic approximation, the Hessian matrix is symmetric and mathematically equivalent to a matrix of second-order partial derivatives of an unperturbed effective harmonic potential energy, $V_{eff}$. Likewise, the Hessian matrix of the perturbation, $H_{p}$, is obtained from the potential energy of the perturbation, $V_{p}$. The elements of the Hessian matrices for the unperturbed system and the perturbation are provided by
(3)$${\left(H_{o}\right)}_{ij}=\frac{\partial^{2}V_{eff}}{\partial x_{i}\partial x_{j}}\;\;\;\;\;\mathrm{and}\;\;\;\;\;{\left(H_{p}\right)}_{ij}=\frac{\partial^{2}V_{p}}{\partial x_{i}\partial x_{j}}\;\;.$$

Notice that Equation (Equation 2) allows the unperturbed Hessian matrix elements in Equation (Equation 3) to be calculated without having to specify the unperturbed $V_{eff}$. The spring placed between atoms *i* and *j* represents a harmonic potential provided by $V_{ij}=\frac{k_{s}}{2}{\left|{\vec{r}}_{i}-{\vec{r}}_{j}\right|}^{2}$ where the position vectors are obtained from the reference structure used for structural alignments. All pairwise harmonic potentials that represent the perturbations are added to obtain $V_{p}$. Then, using Equation (Equation 3), the perturbed Hessian matrix is provided by $H=H_{o}+H_{p}$. In this work, $k_{s}=0.1$ kcal/(mol Å^2^); however, as previously shown [49], the results are independent of this value in the linear response limit, where $k_{s}$ is small. These perturbations to the Hessian matrix change the normal mode frequencies, which are obtained using first order perturbation theory. Following the theory of dynamic allostery set out by Cooper and Dryden [22], the change in the binding free energy (ΔΔG) of an endogenous ligand at the active site induced by the binding of an effector molecule can be calculated from the resulting changes in normal mode frequencies.

The *k*-th normal mode has three frequencies of interest: the apo frequency, $\nu_{0}\left(k\right)$, the frequency with a single endogenous ligand bound at the active site, $\nu_{1}\left(k\right)$, and the frequency with both an endogenous ligand and an effector molecule bound, $\nu_{2}\left(k\right)$. The change in binding free energy can be estimated as
(4)$$\Delta \Delta G=-RT\;\sum_{k>6}\ln\left(\frac{\nu_{1}{\left(k\right)}^{2}}{\nu_{0}\left(k\right)\nu_{2}\left(k\right)}\right)$$
where two sets of external springs are used to model the two binding events. A sum over all modes unrelated to rigid body translations and rotations is performed after removing the first six lowest frequency modes. In practice, a sum over several modes is sufficient to identify putative dynamic allostery sites; ΔΔG quickly converges, as high-frequency modes yield little contributions. It should be noted that because this is a coarse-grained model that uses only carbon alpha atoms, and because the perturbation is small, the ΔΔG are generally much smaller than what would be expected physically; however, it is the shape of the variations as a propensity which is being considered here. This method allows us to perform a single MD simulation on an apo structure and then perform a perturbation scan that calculates changes in active site affinity as different regions of the protein experience the simulated effector molecule binding. When local regions of the effector and endogenous ligand overlap, fewer springs are used, avoiding any double counting of springs. In this work, there is no interesting signal caused by this local interference, and this special case is ignored in all subsequent discussions that focus on distal effects.

### 2.4. Comparative Multivariate Analysis

Through a comparative and exploratory analysis, the functional mechanisms within beta-lactamase were extracted by supervised projection pursuit using Supervised Projective Learning for Orthogonal Completeness (SPLOC) [50,51,52]. The state space consisted of the (x,y,z) coordinates of the carbon alpha atoms representing 263 residues, yielding p=789 degrees of freedom. As only substitution mutations were considered, all proteins had an identical state space. All conformations were structurally aligned using JEDi [45]. The input to SPLOC was a collection of data packets, each quantified by the mean position of the carbon alpha atoms and the covariance matrix describing atomic fluctuations about the mean, respectively denoted as $\mu_{r}$ and $\Sigma_{r}$, where *r* is a data packet index. Multiple data packets (called replicas) were constructed by subsampling conformations in a trajectory. The output of SPLOC provided *p* modes defining a complete orthornormal basis set of vectors, where the *k*-th mode is denoted as v(k). It is worth mentioning that the PCA and SPLOC modes are unrelated, and are used in easily recognizable contexts in this work. In Section 4.1, a quantitative comparison of these two modes is shown and discussed.

A feature space was constructed from the mean and standard deviation of the data projected into each mode, such that $\mu_{r}\left(k\right)$ = $v^{T}\left(k\right)$·$\mu_{r}$ and $\sigma_{r}\left(k\right)$ = $v^{T}\left(k\right)$·$\Sigma_{r}$·v(k). The ordered pair $\left(\mu_{r}\left(k\right),\sigma_{r}\left(k\right)\right)$ defines a point for the *r*-th data packet within a mode feature space plane (MFSP) for mode *k*. As there are two emergent properties (mean and standard deviation) per mode, the feature space dimension is 2p. Considering all data packets for functional and nonfunctional systems, statistical measures were defined per MFSP in order to arrive at a single objective function, called mode efficacy, E(k). The SPLOC process is summarized qualitatively in the next section, with complete details provided in [50].

For the *k*-th mode, there are three measures: (1) selection power, S(k); (2) consensus power, C(k); and (3) quality of clustering, Q(k). Selection is a measure of signal-to-noise, consensus is a measure of statistical consistency across the data (including replicas), and cluster quality is a scale-invariant geometrical measure of how data packets cluster. All these measures are invariant upon the exchange of class labels, which are called functional and nonfunctional in this work. A decision tree involving {S(k),C(k),Q(k)} was used to classify the *k*-th mode as discriminant, undetermined, or indifferent, respectively called d-, u-, or i-mode. If any one of the three measures fell below a minimum threshold, the mode was classified as u-mode. Cross-validation was conducted within SPLOC through the consensus measure as the basis set was optimized. Thus, if a mode was not identified as a u-mode, only two cases were possible; (d-mode, i-mode) could be identified when functional and nonfunctional data packets were linearly separable and statistically indistinguishable within an MFSP. Each mode was assigned a rectifying adaptive nonlinear unit (RANU) in order to bifurcate mode identity towards either d-mode or i-mode. As the parameters of a RANU are data-driven based on the properties of its mode projection, a heterogeneous network of *p* RANUs can determines the mode efficacy, E(k)∀k.

The total efficacy of a proposed orthonormal basis set is provided by $E_{net}=\sum_{k}E\left(k\right)$. As the total efficacy of the network of RANU-type activation functions is linearly separable, the subspaces of the modes can be optimized independently. As such, a generalized Jacobi method was implemented, in which pairs of modes were rotated in different pair combinations through a random search process. During this process, mode efficacy can only increase monotonically, and the algorithm continues to rotate the basis vectors until the total efficacy of the network saturates. As each mode provides a means of determining emergent properties (as means and standard deviation) along the projection, and no information is lost due to completeness, this process provides interpretability of functional dynamics.

### 2.5. Comparative Studies and Data Packet Generation

In this study, two main categories were considered in the course of elucidating the functional dynamics of beta-lactamase by quantifying similarities and differences in dynamics using SPLOC. Each category represents functional versus nonfunctional classes. The first category provides a protein-centric comparison, where differences and similarities in protein dynamics depend on NSBL versus ESBL. The second category provides a ligand-centric comparison, where differences and similarities in protein dynamics depend on broad versus extended spectrum antibiotics. While the protein-centric perspective takes into account all MD trajectories, the ligand-centric perspective is only relevant for proteins when ligands are present. The data packet statistics used in subsequent analyses are summarized in Table 1.

A bootstrap approach was used in which all trajectories that shared a common characteristic were combined. From a given trajectory pool of $N_{traj}$ trajectories consisting of *M* frames each, a subsample of *m* frames were randomly drawn and concatenated together. Through this method, $N_{DP}$ replica data packets for functional and nonfunctional classes and for both categories were constructed. All data packets had $N_{S}=m*N_{traj}$ samples and an observations per variable (OPV) of $N_{S}/p$. When a functional or nonfunctional class consisted of different subclasses (e.g., TEM-1 and TEM-2 are both NSBL), replica data packets were constructed exclusively with MD frames from each subclass. This subsampling process allowed for data packets that provided unbiased representation of all data while at the same time characterizing any fluctuations that might exist.

### 2.6. Extracting Functional Motions

A common measure obtained from MD simulations is the root mean square fluctuations (RMSF). It is common to project out small amplitude motions using a small number of PCA-modes (with largest variances) compared to *p*. In such cases, the actual RMSF and the PCA-projected RMSF will generally be very similar, which is the desired result because it is of interest to capture most of the dynamical motion using a relatively small number of PCA-modes. Using the same projection concept, the RMSF measure can be generalized using the d-modes and i-modes obtained from SPLOC. Specifically, it is possible to observe distinctly different conserved motions between two systems by using (d-modes, i-modes) to project out unwanted motions. When comparing functional versus nonfunctional systems, the observed differences in motion may not be related to function, and conserved motions can be important for function.

By knowing which parts of motion differ between systems and which parts are similar, it is possible to glean insights into which motions support a function and why. A hypothesis involving functional dynamics can then be refined by inductive reasoning as additional comparisons are made. After being projected to be within the (discriminant, indifferent) subspace, dynamical fluctuations are called (discriminant RMSF, indifferent RMSF). The discriminant RMSF (dRMSF) and indifferent RMSF (iRMSF) are calculated using the standard procedure for calculating RMSF, except that the first step is to respectively construct a projection operator from the d-modes and i-modes. The second step then projects out all motions outside the subspace of interest. Where there is a data matrix X(t) having the dimensions p×n and where *n* is the number of frames, the projected data matrices for the discriminant subspace, $X_{d}$, and for the indifferent subspace, $X_{i}$, are provided by
(5)$$X_{d}=\left(\sum_{d\text{-}\mathrm{modes}}v\left(k\right)v{\left(k\right)}^{T}\right)X\;\;\;\;\;\mathrm{and}\;\;\;\;\;X_{i}=\left(\sum_{i\text{-}\mathrm{modes}}v\left(k\right)v{\left(k\right)}^{T}\right)X$$

## 3. Results

### 3.1. Global Motions and Essential Dynamics

Each of the 32 apo and 16 holo trajectories conserved secondary structure across the TEM-1, TEM-2, TEM-10, and TEM-52 beta-lactamase mutants over 500 ns of simulation. In Figure 3, probability densities for the RMSD and essential dynamics are shown in order to quantify the variations in their large scale motions. All MD simulations are pooled together for this analysis. The probability density functions (PDF) for RMSD are shown for each of the four apo and holo mutants, respectively, in Figure 3a,c. Overall, the small RMSD found in all apo and holo structures indicates that the structure of beta-lactamase is very stable. The PDFs for the RMSD in apo structures are generally broader, with an average RMSD around 1.2 Å, while the RMSD for holo structures shows an average RMSD around 1.0 Å. These differences indicate that, on average, beta-lactamase decreases in conformational diversity when a ligand is present in the binding pocket. In apo simulations, the distributions for ESBL mutants are slightly more skewed toward larger RMSD than NSBL mutants; however, the distributions mostly overlap. In the holo form, the distributions are multi-modal, indicating that the enzymes visited multiple conformation states.

Figure 3b,d show the joint distribution for principal components 1 and 2 (a.k.a. PC-1 and PC-2), with the marginal distributions for each component shown on the sides of the plot. in the apo simulations, the essential dynamics in both PC-1 and PC-2 are described as fluctuations around a main single basin. The distributions share significant overlap in PC-1, making PC-1 a poor discriminator between NSBL and ESBL motions. For PC-2, TEM-1 has a greater skew to the right than all other mutants. In the holo simulations, PC-1 and PC-2 describe various sub-populations within the enzyme dynamics. TEM-52, in particular, shows two clear peaks, indicating that PC-1 captures a structural shift when TEM-52 binds a ligand. PC-2 has less overlap between all mutants, while TEM-1 and TEM-52 shift to the left and TEM-2 and TEM-10 shift to the right.

The dynamics of beta-lactamase are fairly constrained, with one or two distinct basins visited by each mutant during the simulation time window. While the dynamics of the apo and holo enzymes span similar phase spaces as defined by PC-1 and PC-2, there is a slight shift of means along PC-1 between the two forms of the enzyme. Based on this analysis, it can be seen that ligand binding reduces flexibility and conformational diversity, while large-scale motions are similar between apo and holo simulations. However, differences occur on a local scale, which requires a more detailed method of extracting the functional dynamics.

### 3.2. Dynamic Allostery

A perturbation scan of the ΔΔG was performed for a putative dynamic allostery binding event at all residues of the TEM-1, TEM-2, TEM-10, and TEM-52 mutants. The results of all four perturbation scans are summarized in Figure 4. As a control, the analysis was applied to each of the eight MD trajectories that represent a single mutant (data not shown). In principle, identical results should be achieved. Although the shape of the response curves is markedly conserved, they differ by different offsets. The same effect is observed for each mutation when all eight MD trajectories are pooled. In order to provide a consistent propensity scale, the allosteric response curves were therefore mean-centered to remove the offset. The average offset for the eight replicates of TEM-1, TEM-2, TEM-10, and TEM-52 was, respectively, 0.0173, 0.0204, 0.0234, and 0.0372 (all positive and in units of RT). The origin of these offsets is likely due to incomplete sampling of the phase space from each MD simulation. With offsets, the allosteric response curve for ΔΔG, as shown in Figure 4a, is markedly consistent across mutants, and common trends are easy to identify, as shown in Figure 4b.

Most strikingly, all three mutants show a similar pattern of allostery response, with the largest conserved shift occurring in the H10-H11 loop. Figure 4b shows that this helix has an accompanying ΔΔG shift on the C-terminal helix, which indicates that this region is an allosteric target. This region has previously been experimentally identified in the literature [53]; Horn et al. showed that these two helices can open to form a cryptic pocket that enables non-substrate molecules to bind non-competitively. Horn attributed a loss of binding affinity as being due to allosteric regulation by the binding partner, forcing Arg244 into a catalytically incompetent conformation. Furthermore, it has been shown that destabilization of this region by disrupting an aromatic ring stacking interaction between Pro226-Trp229-Pro252 residues leads to a total loss of enzymatic function.

In addition to predicting this known allosteric pocket, the perturbation scan predicted multiple other locations for off-target binding sites. Notably, H3 and H4 (residues 99-101 and 110-114, both 21 Å from the catalytic Ser70) shows an inhibiting allosteric regulation effect when constrained, and the H7 helix (residues 150-155, which are roughly 18.5 Å from the catalytic Ser70) shows an enhancing allosteric regulation when constrained. While the perturbation response at these regions is weaker than from the primary allosteric pocket, these results suggest that beta-lactamase can be dynamically modulated by off-binding sites.

#### 3.2.1. Narrow Versus Extended Spectrum Apo Comparisons

Figure 5 shows the respective dRMSF for pairwise comparisons of TEM-2 (Figure 5a,b), TEM-10 (Figure 5c,d), and TEM-52 (Figure 5e,f) mutants versus wild-type TEM-1 in apo form. In comparing dynamic changes that occurred in the ESBL mutants TEM-10 and TEM-52 and not in the NSBL mutant TEM-2, the omega loop (residues 163-178) and the start of H2 (including the catalytic Ser70) experience the greatest change. Overall, both ESBL enzymes showed larger fluctuations in their discriminant motions, reaching up to 0.8 Å compared to 0.3 Å for TEM-2, suggesting that mutations conferring extended spectrum resistance have a greater impact on enzyme dynamics. It is noteworthy that all three mutants exhibit a large discriminant motion at the H9-H10 loop (residues 194-201). An increase in conformational fluctuations in all three mutants, regardless of the specificity of the enzyme substrate, suggests that this loop is important for protein stability.

It can be observed that the dynamic response to the Gln39Lys mutation in TEM-2 is localized to specific residues that are distant from the mutation itself. The largest dynamic differences occur at Glu197, Leu198, Lys215, and Arg241. The closest major change in dynamics to the mutation is 15.4 Å away at Arg241, which is located on the β5-β6 turn close to the omega loop.

The dynamic differences extracted from TEM-10 versus TEM-1 and TEM-52 versus TEM-1 are significantly similar in H2, the omega loop, and the H9-H10 loop. To a lesser extent, dynamic differences are similar at the H10-H11 loop, the β5-β6 turn, and the start of the C-terminal helix. Dynamical differences with respect to TEM-1 are most prominent in TEM-52, with further significant regions in the H3-H4 loop (residues Tyr105 and Ser106) and residues 154-160. The dynamical differences at the H3-H4 loop and the beta turn are likely influenced by the Glu104Lys and Gly238Ser mutations due to local effects.

These results show that the ESBL mutations cause changes at the active site and in the omega loop, as well as weaker long-range effects. The most notable such weaker response is at the H9-H10 loop (residues 194-202), which is distant from both the active site and all three sets of mutations. This relatively hydrophobic loop was the only part of the enzyme that had sensitive dynamic changes to all three sets of mutations. A site-directed mutation study carried out in 1992 [54] showed that TEM beta-lactamase function was resistant to mutations at this linker, with the exception of Leu199. A later study in 2009 [55,56] showed that the Leu201Pro mutation played a stabilizing effect similar to Met182Thr, although with different specificity for stabilizing primary mutations. Other regions that showed a long-range response to amino acid substitutions include the H10-H11 loop and the loop consisting of residues 154-160.

#### 3.2.2. Apo Versus Holo Comparisons

In order to investigate the changes in dynamics before and after ligand binding, all 48 simulations were partitioned into groups by the mutant they represent and then further divided into apo/holo classes for SPLOC comparison (see Table 1 for a summary of the statistics of data packets and how they were generated). As the holo data packets contain an equal number of conformations from the protein that interacts with all four ligands, the covariance matrices discern general binding motions rather than specific ligand motions. Ten replicate runs were performed, with subspace similarity between discriminant and indifferent subspaces showing that SPLOC converged on similar subspace partitions in each run.

Figure 6 shows the dRMSF for each beta-lactamase mutant, indicating where backbone motions were either induced or quenched when a substrate was present in the binding pocket. TEM-1 and TEM-2 (Figure 6a,b) have the greatest response at two locations on the enzyme, while TEM-10 and TEM-52 (Figure 6c,d) show responses that are widely distributed. TEM-1 and TEM-2 show a shift in dynamics in the connecting H9-H10 loop and H10-H11 loop. In TEM-52, the responses occur in loops bordering the binding pocket, including the H3-H4 loop, the H7-H8 loop, and the H10-H11 loop. TEM-10 shows the greatest change in the omega loop and H10-H11 loop.

Upon binding the substrate, the H10-H11 loop had the greatest quench in backbone motion for ESBL enzymes (TEM-10 and TEM-52), while for NSBL enzymes (TEM-1 and TEM-2) the magnitude of motion remained essentially unchanged. This large disparity in the way dynamic changes occur depending on the functional class suggests that this loop plays a key role in coordinating or stabilizing the ligand in the binding pocket for the extended spectrum enzymes. Although the quenching effect on dynamics occurs throughout most of the ESBL enzymes, at the H10-H11 loop it is especially prominent in the TEM-52 mutant.

A close inspection of the MD trajectory revealed that the H10-H11 loop can take several conformations, as illustrated by snapshots in Figure 7a. In the absence of a ligand, conformations of this loop can penetrate into the binding pocket. As shown in Figure 7b,c, the loop tends toward a preferred conformation when a ligand interacts with the enzyme. The extended-spectrum ligands cefotaxime and ceftazidime (shown in cyan) can form a transient contact with the backbone OH of Val216; together with steric exclusion, this could be a mechanism for the increased stability of this loop. The broad-spectrum antibiotics ampicillin and amoxicillin do not reach far enough to form such a contact, and the loop is allowed to flex more, although it nonetheless tends to form a favorable conformation of the ligand. It should be noted that although this interaction helps to stabilize larger ligands, both cefotaxime and ceftazidime were able to remain in the pocket when not constrained to this loop, indicating this is not the primary mechanism by which the ligand is held to the protein.

In addition to the role played by the H10-H11 loop, the dRMSF in Figure 6 highlights the locations of induced motions upon binding in the H9-H10 loop for TEM-1 and TEM-2. This is interesting because the loop is not near the binding pocket, and this region shows the opposite behavior in extended spectrum enzymes, where motion is quenched upon ligand binding. This further supports the idea that this loop plays a general role in stabilizing the protein. The origin of the increased flexibility of this loop within TEM-1 and TEM-2 is around residue Gly196, where the loop can switch between two conformations. This loop acts as a hinge for helices 9 and 10 which connects the omega loop and the H10-H11 loop that borders the active site. As the ligands anchor to these spots, the slight change in conformation could be a dynamic response intended to alleviate stress on the enzyme due to the rigidity of these ligand-interacting loops. This effect is thus weakened when the active site is more accommodating of diverse ligands, as in the ESBL enzymes.

#### 3.2.3. Holo Versus Holo Comparisons

The biochemical distinction between beta-lactamase mutants is in how effectively they bind to a particular ligand. From this perspective, the four beta-lactamase mutants compared here are all functional. The holo simulations represent the various ways beta-lactamase can function in binding either broad-spectrum or extended-spectrum antibiotics. Similarities between these complexes can provide insights into a common ligand-independent inhibition mechanism. Therefore, the holo simulations were divided and compared according to both the protein perspective (NSBL versus ESBL) and the ligand perspective (broad-spectrum versus extended-spectrum antibiotics). From the protein perspective, SPLOC found 316±31 d-modes and 407±41 i-modes. For the ligand perspective, SPLOC found 241±21 d-modes and 499±28 i-modes. The most prominent differences in discriminant fluctuations were that the protein perspective detected greater motion in the omega loop. Based on the analysis of the apo trajectories, this difference is likely related to the dynamical changes induced by ESBL mutations.

In the indifference subspace, molecular motions are statistically indistinguishable. The iRMSF from the ligand perspective, shown in Figure 8a, indicates areas where motion is conserved, regardless of which ligand interacts with it; these regions include the omega loop, H9-H10 loop, and H10-H11 loop noted above. As shown in Figure 8b, many regions are far from the active site, such as residues 50-55, 154-160, and 194-199, which are flexible loops on the opposite side of the protein in relation to the active site.

## 4. Discussion

TEM beta-lactamase is dynamically constrained on a global scale while exhibiting variations in loop motions sensitive to mutations and ligand binding. In agreement with other recent studies [25,57], the present study further reveals the importance of loop dynamics in controlling beta-lactamase function. In particular, the H9-H10 loop facilitates structural stability, and the H10-H11 loop stabilizes the ligand in the active site for catalysis. Moreover, different long-range effects between the various mutants provide a means for adaptability in beta-lactamase, promoting further control of substrate specificity.

### 4.1. Comparison of PCA and SPLOC Modes

Dimension reduction using PCA to describe essential dynamics has long been used as a standard method for extracting functional motions from molecular dynamics simulation data [58,59]. Despite its popularity, PCA captures functional dynamics that correspond to the largest amplitude motions described by collective modes with the greatest variance. However, this approach is actually based on an assumption that is not universally true; function can involve motions in localized regions that are not extended. The usual practice of throwing out the low-variance PCA modes thus risks missing the functionally important motions altogether. In the analysis provided herein, PCA does not successfully cluster known functional data classes, as shown in Figure 3. In this case, a supervised discriminant analysis method is more appropriate. Due to the data-driven efficacy score of SPLOC, motions can be labeled as discriminant or indifferent regardless of the amplitude of motion. It is thus interesting to determine how much information contained in orthogonal subspaces of SPLOC can be reconstructed with PCA modes ordered from the most to the least variance.

In Figure 9, the SPLOC modes from the results in Section 3.2.1 are compared with the global PCA modes obtained in Section 3.1 in order to determine how the information extracted by SPLOC is distributed across the PCA basis. Each basis vector in a given subspace was reconstructed using the top *n* PCA modes and averaged over all modes in the SPLOC subspace for both SPLOC subspaces (comprising only d-modes, i-modes, and u-modes) and the full SPLOC basis set (Figure 9a–d, respectively). The percentage of the reconstructed mode is the cumulative overlap (CO) between the SPLOC basis vector and the PCA subspace, provided by
(6)$$CO\left(\vec{v},U\right)=\sum_{\vec{u_{i}}\in U}{\left(\vec{v}\cdot \vec{u_{i}}\right)}^{2}\;\;\;.$$

Figure 9a shows that, on average, the d-modes were not even 20 percent reconstructed using the top 400 PCA modes. This means that information about the differences in functional dynamics between two beta lactamase mutants is lost due to dimension reduction. The i-mode reconstruction shown in Figure 9b increases linearly in CO until 500-600 PCA modes are used. The u-mode reconstruction shown in Figure 9c initially increases rapidly, especially compared to TEM-1 and TEM-2. Finally, if we use the complete set of SPLOC modes, a perfect linear relationship emerges with a slope of 1/789, as seen in Figure 9d. Therefore, it makes sense to compare the reconstruction rate with the reference line shown in Figure 9d.

The i-modes describe the functional similarity between mutant pairs that is most conserved. This motion is expected to be associated with the dominant enzyme dynamics through large-scale motions, which are mainly determined by the structure of the protein. Due to the enzymes’ high degree of structural similarity, the CO of i-modes expressed in terms of PCA modes increases faster than the reference line until the dimension of the i-mode subspace is reached. The motions described by u-modes are not statistically different or the same between a pair of mutants, and thus capture outlier motions that may have been expressed differently between simulations of the same enzyme. Being sensitive to outliers, PCA often captures this high variation in the top modes, which explains why the u-modes initially increase rapidly in CO until the dimension of the subspace for u-modes is reached. Finally, Figure 9a clearly shows the statistically significant trend that d-modes contain very different information from the most important PCA modes. This analysis suggests that the SPLOC analysis provides a means of capturing different aspects of protein motions that are important for function that PCA cannot provide.

### 4.2. The H9-H10 Linker Loop Stabilizes Beta-Lactamase against Structural Changes

The MD/SPLOC results suggest that the H9-H10 loop plays a key stabilizing role, which is in agreement with previous work [57] postulating that loops transmit allosteric signals through the H9-H10 loop, resulting in the observed high fluctuations. This loop (shown in Figure 10) connects two helices that are almost perpendicular to each other. The H9 helix at the beginning of the loop shields the back of the H2 helix and the catalytic serine 70 from solvent. This loop sterically counters a resultant push back of the H2 loop when it binds to an antibiotic. From the simulation data, Leu194 braces Leu51 on the nearby β1-β2 connector loop. Mutations such as Leu51Pro, Leu51Phe, or Leu51Ile (e.g., in TEM-60) have been shown to increase the stability of TEM-1 beta-lactamase [17,60], which supports this hypothesis. Additionally, residue Lys192 has been observed to form an intermittent interaction with Gly196, and disruption of this favorable interaction should provide the means for greater flexibility.

The results in Section 3.2.1 show that both specificity-modulating and functionally-silent mutations can induce a dynamic response in the H9-H10 loop. Notably, all three mutation sets were able to induce at least a small change in the top of the omega loop, which directly connects to the H9 loop. The smallest change in the omega loop dynamics was observed in TEM-2. In Section 3.2.2, it was shown that this loop remains flexible when NSBL enzymes bind to a drug, whereas it becomes more rigid when ESBL enzymes bind to a drug. This is in response to the flexibility induced in the omega loop and active site by ESBL-conferring mutations. The active site takes the more flexible form when it accommodates a ligand, as it takes up slack from the H9-H10 loop. Furthermore, this loop is an intermediate connector between the omega loop and the H10-H11 loop, which forms the top and bottom of the catalytic pocket. These three loops are all key locations where the ligand can anchor itself to the enzyme and where flexibility is altered by mutation or effector binding.

### 4.3. The Role of the H10-H11 Loop in Stabilizing Ligands

Extended spectrum ligands such as cefotaxime and ceftazidime are typically larger than their broad spectrum counterparts, and are therefore harder to accommodate in the active site. Analysis comparing all three mutation sets shows a change in motion in the H10-H11 loop for TEM-10 and TEM-52 (Figure 5c,d). A comparison of apo and holo enzymes shows that ESBL enzymes lose flexibility here, while NSBL enzymes exhibit similar dynamics before and after the ligand enters the pocket. Comparing the dynamics in the holo simulations (Figure 8a) shows that although the loop motions change upon binding, they remain relatively mobile. As shown in Figure 7, the carboxyl group of cefotaxime and ceftazidime points toward the H10-H11 loop, as does a carbonyl group in cefotaxime and pyridine group in ceftazidime. These functional groups form transient contacts with Val216, which stabilizes the ligand, beyond the primary anchors of the ligand at the β5-β6 turn and the omega loop. The conformational freedom of the H10-H11 loop enables the ligand to move towards the active site and positions Val216 near the carboxyl group of the ligand. This underscores the importance for ligand stability of having Val216 in the correct conformation, which explains why motions drastically change upon binding to an extended-spectrum ligand.

Contact with the H10-H11 loop does not appear necessary for ligand hydrolysis, rather promoting the capture, chaperoning, and stabilization of the protein ligand complex. For example, cefotaxime was able to resist being pulled into the solvent, although this H10-H11 loop contact was only intermittent during the simulation. In the TEM-1 holo simulation, the cefotaxime was able to reposition itself in the pocket to interact with the H10-H11 loop while remaining anchored to the omega loop. Ampicillin and amoxicillin can reside in the binding pocket, despite not being able to reach this loop. This secondary anchor forces the extended spectrum ligands to stretch across the active site, putting them in a more favorable position for catalysis. Although not required for function, these results show that this loop plays an important role in expanding substrate specificity for TEM beta-lactamase.

### 4.4. Dynamic Allostery as a Mechanism of Inhibition

The dynamic allostery perturbation scans of the four TEM beta-lactamase mutants suggest that substrate binding is coupled to distal loop regions. In combination with previous results from the literature, a mechanism involving the known allosteric binding pocket for reducing antibiotic affinity is examined here in detail.

According to the scan, binding an effector to H11 is likely to enhance the binding affinity of antibiotics at the active site. In prior experimental work, destabilization of this region was shown to inhibit activity [53,61]; here, a stabilizing perturbation was found to enhance activity. The mechanism for H10-H11 ligand stabilization above relies on Val216 taking the appropriate conformation to reach the ligand. A plausible explanation of this allosteric response would be that removing flexibility in this region induces the H10-H11 loop into a favorable conformation for ligand binding. By contrast, forcing H11 and H12 away from each other is likely to move the loop away from its favorable conformation, thereby reducing the enzyme’s activity.

Binding an effector near residues 227-229 and nearby to residues 254-256 may inhibit binding of ligands at the active site. This mechanism is directly related to the primary allosteric region of beta-lactamase, as it houses the pi-stacking Pro226-Trp252-Pro229 triad, which is thought to stabilize this region. Strain in this region due to effector binding increases the likelihood of disrupting this interaction and further disrupting the structure of the active site, including the H10-H11 loop via long-range dynamic signal propagation. SPLOC identified large conformational fluctuations in this loop, enabling the enzyme to adapt the shape of the binding pocket to various beta-lactam molecules.

Another allosteric target area was found at residues 104-106 and 111-115. This allosteric signal is strong for all mutants; however, in TEM-52 this reflects the location of the Glu104Lys mutation. Tyr105 on this loop has been previously identified as an important residue via mutation studies [62] and NMR experiments [63]. Mutation study showed that this position prefers an aromatic residue to cap the active site and prevent steric interactions with the ligand in the pocket. Moreover, NMR showed that Tyr105 dynamics are correlated with both the motions of other local active site residues and with distant residues. These residues include Asn132, a catalytic residue found on the loop just below Tyr105, and Lys234, found on the β5-β6 loop on the opposite side of the active site. The same study found a correlation between the motions of Tyr105 and Val216, suggesting a link between the allosteric mechanism described above and the H3-H4 loop. This work further suggests that this allosteric signal can be controlled to inhibit the enzyme by binding events at either H3 or H4, which form the beginning and end of this loop.

It appears that the H9-H10 loop resists structural changes to beta-lactamase, although it was not identified as allosteric in the perturbation scan. While loop dynamics may not directly couple to active site residues, they are influenced by distant residues. Therefore, the H9-H10 loop may be a viable inhibitor target, as its flexibility is well conserved across mutation sets and ligand binding differences in TEM beta-lactamase. As an alternative approach to dynamic allostery, SPLOC results can be used to gain insights into how to optimize dynamic properties that improve enzyme binding. Binding a ligand to or near this loop may force the H9-H10 loop into a conformation with more rigidity, preventing the loop from maintaining stability through its dynamic mechanism. The region between H10 and the β6-β7 turn, shown in Figure 11, forms a pocket roughly 17.6 Å long by 15.0 Å wide.

## 5. Conclusions

The functional dynamics of TEM beta-lactamase for substrate recognition were investigated using MD simulations. Perturbation scans revealed allosteric target sites, including the known allostery pocket between the H11 and H12 helix and the H3-H4 helix–loop region. Multivariate comparative analyses applied to the simulation data provided a detailed look at the way in which point mutations and ligand binding affect the internal motion of the protein. These comparisons showed that certain mutations from TEM-1 introduce flexibility into the active site, allowing extended-spectrum antibiotics of various shapes to fit into the binding pocket.

The dynamics in most loops contribute to the way in which the protein responds to ligand binding and mutations. Comparing different mutants with the wild type TEM-1 enzyme in apo form revealed that mutations modulate the flexibility of the active site, which influences specificity. The ligand is primarily anchored in the binding pocket by the omega loop and the β5-β6 turn. For larger ligands, the molecule can transiently form contacts with the H10-H11 loop at the base of the active site in order to stabilize the antibiotics from being pulled into the solvent. This loop mitigates the primary allosteric response of beta-lactamase when the interactions stabilizing contacts between H11 and H12 are broken. A novel function-modulating site at the H9-H10 loop is proposed, with Leu51 thought to maintain local structure while the loop reduces strain in the protein. More generally, the methods used to quantify functional dynamics can inform future work on the design of inhibitors intended to combat antibiotic resistance by controlling stress response.

## Figures and Tables

**Figure 1 entropy-24-00729-f001:**
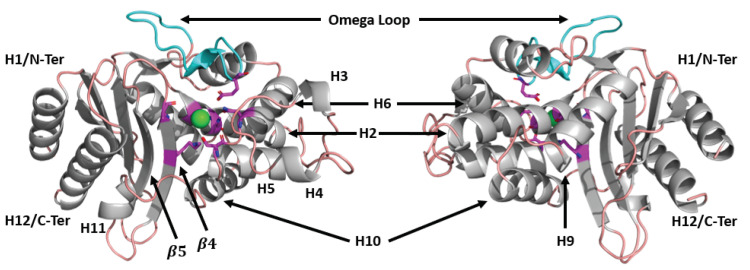
TEM beta-lactamase, labeled with important structural features. Magenta indicates residues involved with catalysis, including Ser70, Lys73, Ser130, Asn132, Glu166, Lys234, and Ala237. The primary binding site for beta-lactams is shown as a green sphere. Loops are shown in light pink, and cyan indicates the location of the omega loop (residues 163-178).

**Figure 2 entropy-24-00729-f002:**
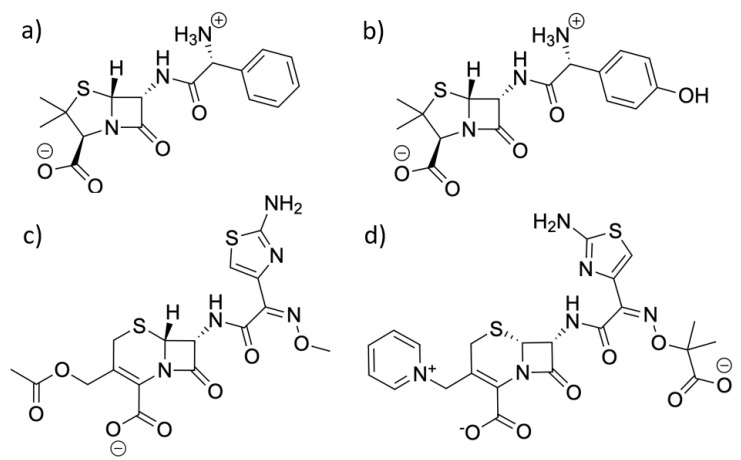
Structures of the four ligands used for simulation in this work: (**a**) ampicillin, (**b**) amoxicillin, (**c**) cefotaxime, and (**d**) ceftazidime.

**Figure 3 entropy-24-00729-f003:**
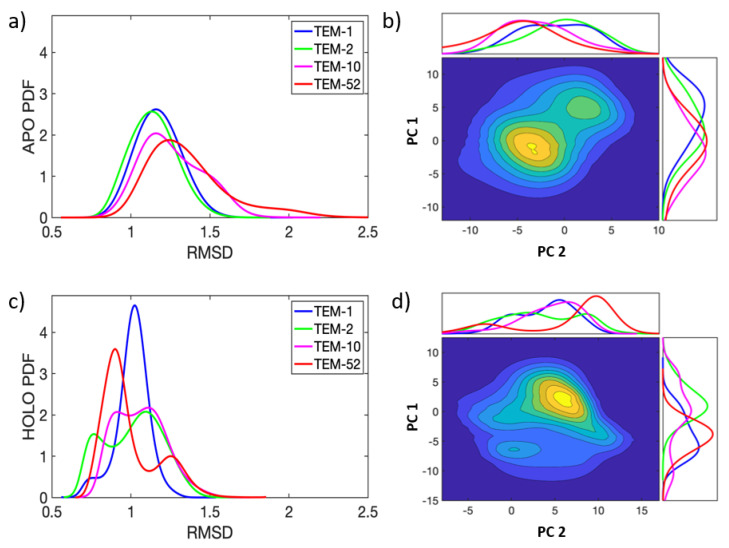
Essential dynamics for apo (top row) and holo (bottom row) TEM beta-lactamase. The RMSD (**a**,**c**) and PC projections (**b**,**d**) have units of Å. The heat maps were calculated by pooling all MD simulations together. Variations can be seen in the individual PDF plots along the *x*-axis and *y*-axis for the apo/holo heat maps, which use the same coloring as in (**a**,**c**).

**Figure 4 entropy-24-00729-f004:**
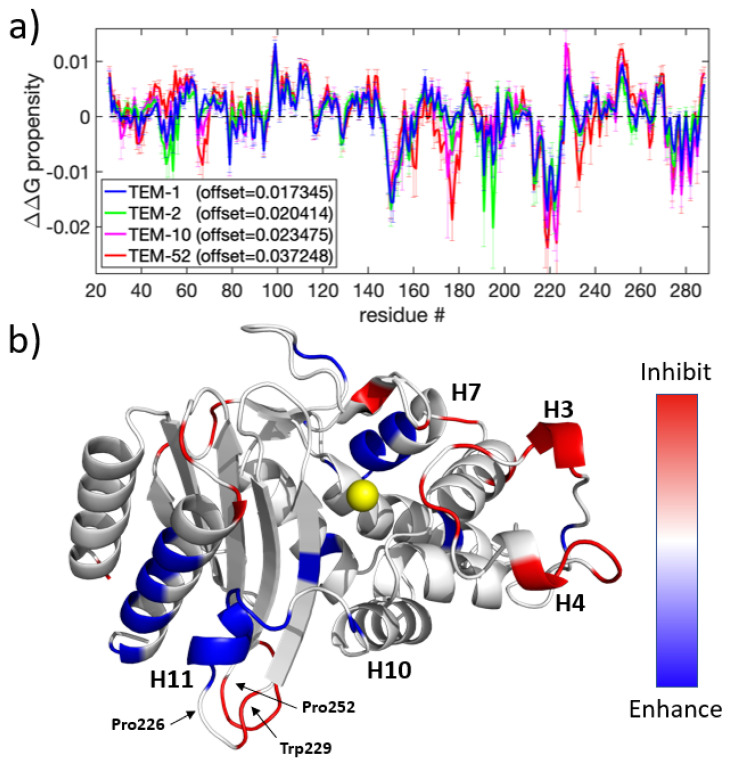
Allosteric targets: (**a**) The ΔΔG propensity response for TEM-1, TEM-2, TEM-10, and TEM-52; (**b**) a 3D rendering of the same data shown in (**a**), with (blue, red) showing areas where rigidifying perturbations (enhance, inhibit) ligand binding. Because the general trend is similar across all four mutants, the average signal for all four mutants displays the general allosteric regions of beta-lactamase. Signals less than 0.0025 are zeroed in order to show better contrast on sites with a higher signal.

**Figure 5 entropy-24-00729-f005:**
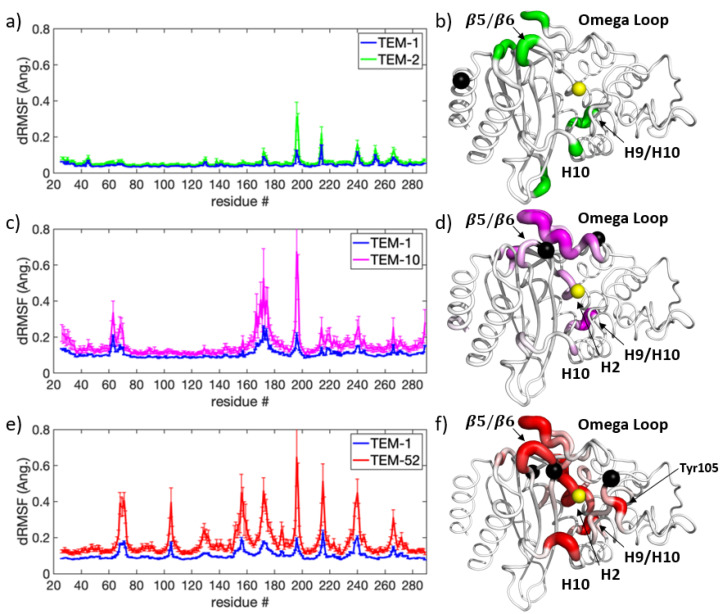
Dynamical differences through dRMSF. Using TEM-1 as a baseline reference, dynamical differences are shown using dRMSF in contrast to mutant (**a**) TEM-2, (**b**), TEM-10 and (**c**) TEM-52. Corresponding 3D renderings using pymol are shown in (**d**–**f**) using the same coloring scheme. On all structures, the yellow sphere represents catalytic Ser70 for reference and the black spheres show the location of each mutant’s amino acid substitution locations.

**Figure 6 entropy-24-00729-f006:**
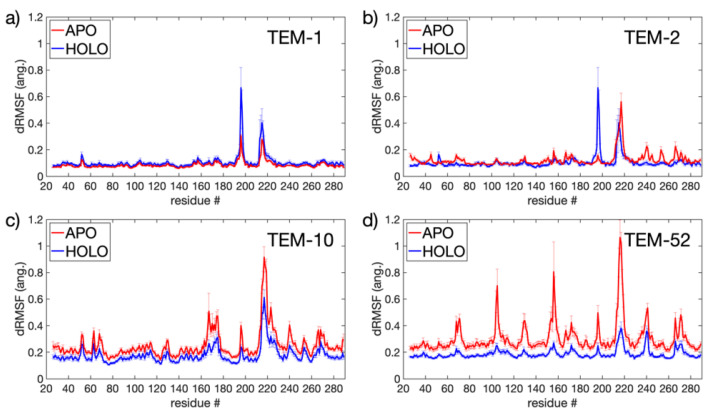
dRSMF for apo vs. holo simulations for: (**a**) TEM-1; (**b**) TEM-2; (**c**) TEM-10; and (**d**) TEM-52.

**Figure 7 entropy-24-00729-f007:**
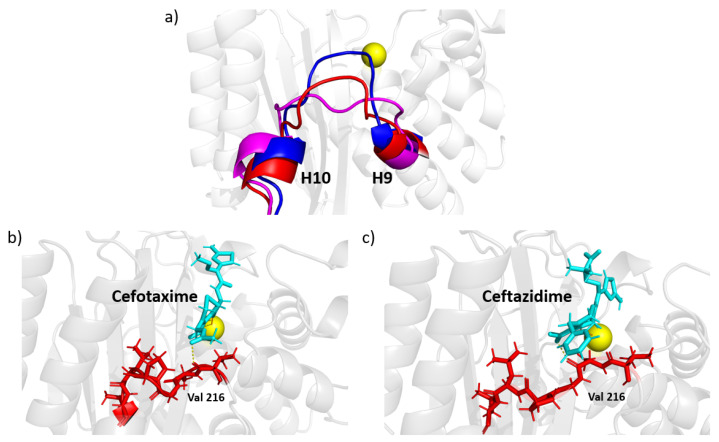
H10-H11 loop conformations. (**a**) Selected example conformations observed in apo simulations. The carboxyl groups of the extended-spectrum ligands cefotaxime (**b**) and ceftazidime (**c**), shown in cyan, are large enough to reach Val216 and form a stabilizing contact.

**Figure 8 entropy-24-00729-f008:**
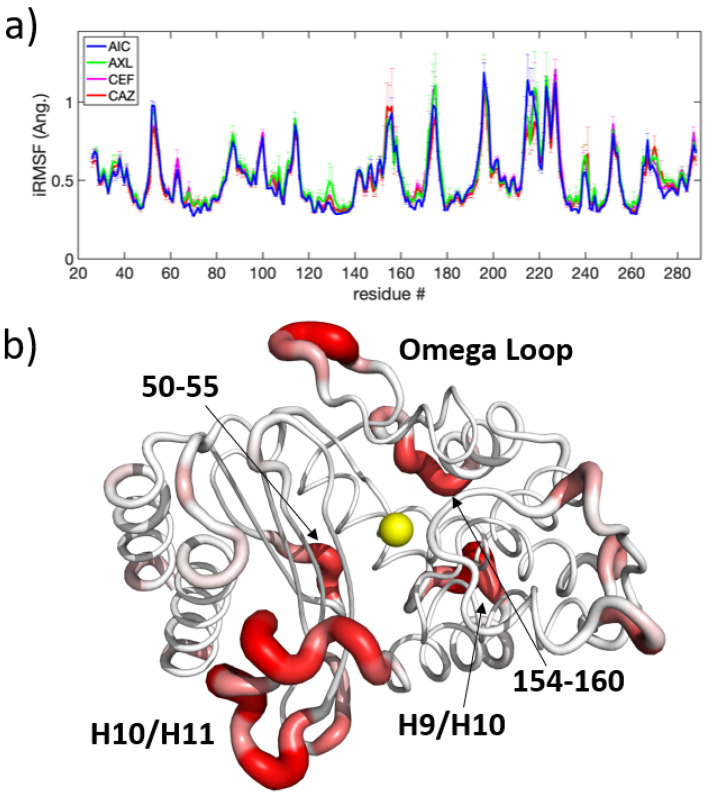
Conserved dynamics: (**a**) iRMSF for holo trajectories in the ligand perspective; (**b**) average iRMSF rendered onto a beta-lactamase structure, with thresholding to emphasize conserved regions.

**Figure 9 entropy-24-00729-f009:**
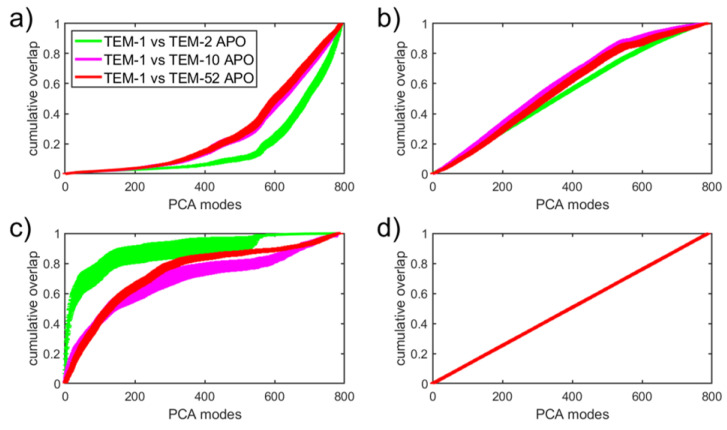
The average cumulative overlap (CO) between SPLOC modes and PCA modes in (**a**) discriminant space, (**b**) indifferent space, (**c**) undetermined space, and (**d**) over the entire basis set. CO is computed for each mode in the SPLOC subspace by summing over the top *n* PCA vectors. The results shown here are the average over all replicate SPLOC runs and all modes within each subspace, with the error bars representing the standard error.

**Figure 10 entropy-24-00729-f010:**
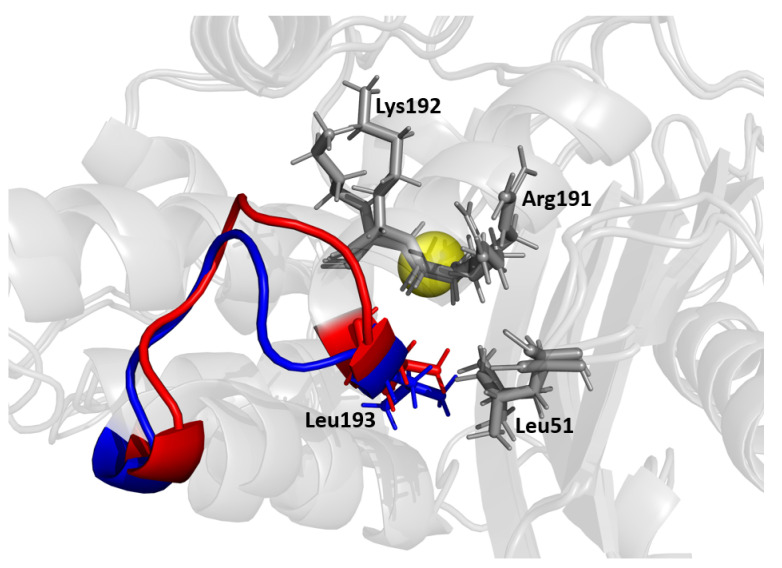
Multiple conformations reflecting the dynamic motions of the H9-H10 loop. Residues Arg191, Lys192, and Val51 are shown as potential residues that influence this region. The active site Ser70 is shown as a yellow sphere for reference.

**Figure 11 entropy-24-00729-f011:**
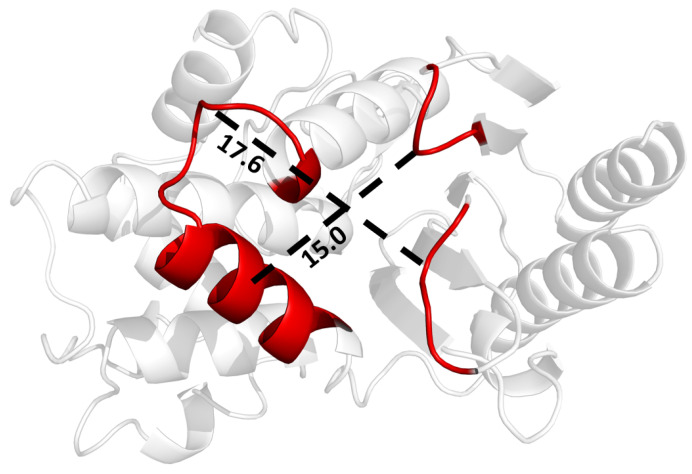
Potential binding pocket on beta-lactamase for inhibiting motions in the H9-H10 helix, leading to enzyme destabilization as it tries to bind a ligand.

**Table 1 entropy-24-00729-t001:** Description of data packet statistics for each class of trajectories compared. In the protein category, NSBL is compared to ESBL, while in the ligand category, broad spectrum ligands are compared to extended spectrum ligands; $N_{traj}$ is the number of trajectories in the pool, *M* is the number of frames per trajectory, *m* is the number of frames that are randomly subsampled, $N_{DP}$ is the number of data packets constructed, $N_{S}$ is the total number of frames per data packet, and OPV is the number of observations per variable in each data packet.

	$N_{\mathrm{traj}}$	*M*	*m*	$N_{\mathrm{DP}}$	$N_{S}$	OPV
apo protein category	8	8000	4000	16	32,000	40.55
holo protein category	4	8000	4000	16	16,000	20.25
holo ligand category	4	8000	4000	16	16,000	20.25

## Data Availability

The data presented in this study are openly available in https://zenodo.org/record/6565237#.YocFZXbMJPZ (accessed on 4 May 2022), https://zenodo.org/record/3523213#.YockzFRByUk (accessed on 4 May 2022).

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
