# Peer review of "Functional Dynamics of Substrate Recognition in TEM Beta-Lactamase"

_entropy, 2022, doi:10.3390/e24050729_

Round 1

Reviewer 1 Report

This is a very interesting and rigorous work that allows for a comprehensive analysis of different mutants dynamics and to highlight the bases of allosteric phenomenon.

I have only (small) methodological issue, the authors write:

The output 216 of SPLOC gives p modes defining a complete orthonormal basis set of vectors, where the 217 k-th mode is denoted as v(k). It is worth mentioning that the PCA and SPLOC modes are 218 unrelated, and are used in easily recognizable contexts in this work.

This is not strange given the supervised character of SPLOC with respect to PCA (similar behaviour is observed in the comparison between PLS and PCA) but, at odds with PLS, SPLOC is a much less common technique and needs further specification (referring to articles is not sufficient) especially as for the Y variables that drive the generation of SPLOC modes. Moreover the article will benefit by a canonical correlation analysis comparing SPLOC and PCA spaces so to get a global understanding of the coherence of the two spaces. The fact that both SPLOC and PCA modes are independent (within their own space) make the canonical analysis well-conditioned eliminating spurious ‘within group’ correlations (see Sirabella, P., Giuliani, A., Colosimo, A., & Dippner, J. W. (2001). Breaking down the climate effects on cod recruitment by principal component analysis and canonical correlation. Marine Ecology Progress Series216, 213-222.).

After this addition the manuscript deserves publication.

Author Response

Thank you for your positive comments and suggestions regarding showing how different PCA modes are compared to SPLOC modes. We performed Canonical Correlation Analysis using SPLOC and PCA modes on our MD trajectories to show that PCA and SPLOC i-modes retain information in the trajectories which allow for high correlations (>.86) to be extracted, while d-modes removed such information reducing canonical correlations to <.72. These results suggest that i-modes and PCA modes captured the related correlated motions in the trajectories while the d-modes removed major correlations. However, reporting these correlations is not as clear as a direct reconstruction of SPLOC modes in terms of PCA modes.

We decided to take the opportunity to demonstrate that there is a difference between the SPLOC d-modes and the top PCA modes. We were not surprised by this result because we have been dealing with both SPLOC and PCA on this system for some time. However, your suggestion pointed to an important point that should be emphasized. As such, we have added another subsection in the Discussion section. We have also added two references that discuss the general approach of essential dynamics in terms of PCA modes. This change is marked in the uploaded PDF in red text. In the latex version, we do not color the changes.

Some minor change (shown in red text) was made in the method section, where the reviewer spotted the note that we made; stating that SPLOC modes and PCA modes need not be the same. We forecast that we do a direct comparison in the Discussion.

Now in Figure 9 (a new figure), we explicitly show that d-mode, i-mode and u-mode subspace reconstructions with PCA modes have different properties and discuss these differences.

Reviewer 2 Report

The article by Avery et al shows the application of the SPLOC  machine learning protocol, recently described by the authors,  to identify functionally important dynamic features in beta lactamases comparing narrow spectrum beta lactamases and extended spectrum beta lactamases in the apo and holo form bound to four  beta-lactam containing ligands.

Extended dynamic runs starting from in silico mutated models based on distinct X-ray structures were analyzed.  This approach, while departing from the experimental models offers and increased sample of the conformational variation. 

Ligands were incorporated by rigid docking in one of the protein variants and then transferred to the other variants.  This is unavoidable because the natural ligands would be hydrolyzed by the natural enzyme.  

Extended molecular dynamics were performed in a temperature and pressure equilibrated TIP3P solvated system with sodium ions to ensure electrical neutrality.  Molecular dynamics was used to generate an ensemble of conformations, although, as far as I understood, the time correlation was not used and sub ensembles were generated by random sampling along the trajectories.

 In addition to the comparative multivariate analysis of the trajectories, the authors explore dynamic allostery by simulating the effect of a perturbing ligand by applying an harmonic potential to atoms located within a 10 A sphere of each of the carbon atoms and computing the perturbation of the Hessian matrix from the second derivative of the potential, added to the empirically determined unperturbed Hessian extracted from the covariance matrix.  This is a clever approach that allowed the identification of potential allosteric sites.

The article is well written and illustrates the potential of the SPLOC approach to identify functional dynamics.  It also describes a straightforward approach to scan for dynamic allosteric sites by introducing position dependent perturbations into a single molecular dynamic simulation.

The article can be published as it is but, if the authors want to make easier to follow the discussion of the results, I recommend marking the position of the secondary structure elements or the relevant regions in the plots where residue numbers are used (fig 4, 5, 6 and 8).

Also, I was intrigued by the sentence around lines 112 stating that “In particular, the reactive oxygen of the beta-lactam ring was positioned so that it could interact with the Ser70 residue on beta-lactamase.”  I did not look at the literature on the beta lactamase mechanism, but I would have guessed that the chemically relevant interaction would be between the oxygen of Ser70 and the carbonyl carbon of the beta-lactam (not the oxygen)

Author Response

Thank you for catching our mistake with regard to the reactive oxygen. Our statement was in error, and indeed we checked this carefully and changed the sentence.

We also took your suggestion and added more labels to the figures that show protein structure.

Round 2

Reviewer 1 Report

The manuscript should be accepted for publication and the authors complimented for a very clear and insightful work.